# Microfluidic Chamber Design for Controlled Droplet Expansion and Coalescence

**DOI:** 10.3390/mi11040394

**Published:** 2020-04-10

**Authors:** Mark Kielpinski, Oliver Walther, Jialan Cao, Thomas Henkel, J. Michael Köhler, G. Alexander Groß

**Affiliations:** 1Microfluidics Group, Institute for Photonic Technologies, IPHT-Jena, Albert-Einstein-Str. 9, 07745 Jena, Germany; thomas.henkel@ipht-jena.de; 2Department of Physical Chemistry and Microreaction Technologies, Institute of Chemistry and Biotechnology, Technische Universität Ilmenau, Prof.-Schmidt-Straße 26, 98693 Ilmenau, Germany; oliver.walther@tu-ilmenau.de (O.W.); jialan.cao@tu-ilmenau.de (J.C.); michael.koehler@tu-ilmenau.de (J.M.K.)

**Keywords:** droplet microfluidics, segmented flow, channel confined, network interface, electrocoalescence, labdisc, self-controlled, compact disk CD-production, volume bridging, bypassed chamber

## Abstract

The defined formation and expansion of droplets are essential operations for droplet-based screening assays. The volumetric expansion of droplets causes a dilution of the ingredients. Dilution is required for the generation of concentration graduation which is mandatory for many different assay protocols. Here, we describe the design of a microfluidic operation unit based on a bypassed chamber and its operation modes. The different operation modes enable the defined formation of sub-µL droplets on the one hand and the expansion of low nL to sub-µL droplets by controlled coalescence on the other. In this way the chamber acts as fluidic interface between two fluidic network parts dimensioned for different droplet volumes. Hence, channel confined droplets of about 30–40 nL from the first network part were expanded to cannel confined droplets of about 500 to about 2500 nL in the second network part. Four different operation modes were realized: (a) flow rate independent droplet formation in a self-controlled way caused by the bypassed chamber design, (b) single droplet expansion mode, (c) multiple droplet expansion mode, and (d) multiple droplet coalescence mode. The last mode was used for the automated coalescence of 12 droplets of about 40 nL volume to produce a highly ordered output sequence with individual droplet volumes of about 500 nL volume. The experimental investigation confirmed a high tolerance of the developed chamber against the variation of key parameters of the dispersed-phase like salt content, pH value and fluid viscosity. The presented fluidic chamber provides a solution for the problem of bridging different droplet volumes in a fluidic network.

## 1. Introduction

During the last decade, droplet-based microfluidics [1] has been established as a very powerful tool for parallelized chemical syntheses [2,3] as well as for biological and biomedical research [4] and various screening applications [4,5,6]. In general, two different techniques are available: (a) emulsion-based droplets processing or (b) segmented flow or channel confined droplet processing. Emulsion-based techniques make use of droplet populations composed of free droplets stabilized by a surfactant. In this way huge number of droplets can be prepared and processed, but the information about the droplet composition at the time of formation gets lost in the emulsion. However, even small emulsion droplets have to be arranged in serial manner and transferred to a channel confined environment for assay readout procedures [7,8,9,10]. Channel confined droplet- or micro segmented flow techniques in a sub-µL volume [11] were developed for the generation and manipulation of droplets in ordered sequences of individually composed droplets with well-defined process history. Each droplet in such a serial sequence can be assigned to its fluidic history, and the composition can be decoded by determination its position within the sequence. The identification of individual droplets due to their formation and process history within a multi-droplet sequence is one of the main advantages of micro segmented-flow techniques compared to the emulsion-based technique, where the droplets arise as a mixed population without spatial confinement. This technique is an important tool to realize experiments in the sub-microliter and nanoliter range that require complex concentration variations [12]. The segmented flow technique has been successfully introduced for a wide spread field of applications such as: (a) concentration-dependent synthesis of different types of nanoparticles [13]; (b) cultivation and susceptibility studies of microorganisms [14,15], in particular, for highly-resolved dose/response functions in micro-toxicology [16,17]; (c) for the screening of two- or three-dimensional concentration spaces and for realizing the principle of stochastic confinement for the search of rare microorganisms [18,19]; and (d) the synthesis of polymeric micro-rods [20], to name some exemplary applications. 

Depending on the assay applications, the droplet volumes vary considerably. Picoliter droplets or droplets of one or a few nanoliters are mainly used for emulsion-based applications [21], but have to be handled in channel confined manner for readout or sorting purposes [22,23,24]. The micro segmented-flow applications in tube or pipe arrangements preferably use droplet volumes in the range between 5 nL and 1 µl [15]. The medium nanoliter level (about 35 nL) is of interest for the application of micro segmented-flow in the cultivation of microorganisms and other cells in chip or microfluidic disc devices [25]. Beside the accurate droplet formation, the controlled transport of droplets, pair-wise droplet fusion [26,27] and droplet splitting [28] as well as the addition of reagents into droplets [29] are essential fluidic tasks for the operability of assays in microfluidic devices. However, there is still a technological gap between these two droplet processing strategies. Emulsion-based techniques are favorable for single-out or cherry-picking strategies using huge droplet populations of tiny droplets. The strictly serial segmented flow technique is favorable, for dose/response screening operations with well-known droplet content in sub-µL volume. However, the combination of both principles needs a fluidic interface which allows appropriate transfer operations on a droplet level. In droplet-based microfluidics, dilution of dissolved compounds means the expansion of an initial droplet volume to a larger volume [30,31]. The well-defined coalescence of groups of droplets with different ingredients or compositions allows on the one hand the procedural conversion of low-volume droplets into larger-volume droplets and on the other hand the stochastically confined combination of different droplet-based building blocks. This operation is not only of interest for combinatorial experiments [32,33,34], or dilutive selection strategies (stochastic confinement), but also bridges the gap between lower and higher volume levels inside microfluidic systems or networks. The expanded/diluted droplets may be compartmentalized back into smaller droplets in a subsequent process step [32]. 

The plurality of biological assays are dose/response type experiments, in which an effect is determined as a function of an effector concentration. Therefore, a defined concentration graduation of the effector in the different samples is required. Hence, the motivation for the development of droplet dilution strategies is the generation of droplet series with concentration graduation. Various droplet dilution strategies have already been described to solve this problem, e.g., the adjustment of the effector concentration during the initial droplet formation [16,35,36], or the volume expansion of the initial droplets by adding a dilution media [11] and the serial dilution of trapped droplets [37].

The use of bypassed channels is a proven strategy for controlling droplet motion in fluidic networks without active elements. A bypass channel equilibrates the pressure between two points of a main channel which are connected by the bypass. Self-controlled operations, triggered by the moving droplets themselves were realized in this way. Only the pressure-driven flow drives the droplet operations in these cases [38,39]. Different examples such as droplet (a) formation [40,41], (b) merging [42,43], (c) splitting [44], (d) sorting [45,46], (e) synchronization [47] and (f) trapping [48,49,50,51,52] have already been described.

Stable emulsion processing without undesired random droplet coalescence of contacting droplets requires surfactants for its stabilization [53,54]. On the other hand, desired droplet coalescence has to be triggered externally and cannot be facilitated by simple droplet collisions. The selective coalescence of neighboring droplets can be achieved by applying electric fields. The so-called electrocoalescence is necessary to break the stabilizing surface energy of the droplets, especially if surfactants are used. Various examples of selective coalescence of droplets were reported earlier [29,30,55,56,57,58,59,60,61,62].

Recently, we have developed an integrated fluidic network for bacterial screening purposes (see Appendix A). The general task of the network is the kinetic investigation of dose/response relationships on different volume scales: 5, 35 and 500 nL. The fluidic network contains different modules with generic operation units providing basic laboratory processes. All modules can be operated either independently or pre- or post-operational of another module. As mentioned before, an essentially necessary assay operation is the defined dilution of droplets, which cause a volume and dimensional expansion. The aim for the bacterial screening is the proliferation of bacteria from selected droplets by adding fresh media, and subsequent analysis their growth kinetic. The dilution of the initial concentration of stimulating effectors can also be achieved by the chamber. The presented operation unit presented was designed to solve these issues (Figure 1). The unit layout details as well as its function and the different operation modes are described here.

## 2. Experimental and Setup

The bypassed chamber device was designed for liquid/liquid two-phase applications using aqueous and fluorinated phases stabilized by fluorinated surfactants [54]. The geometries of the channels were adapted to input droplet volumes in the mid-nL range, preferably of single volumes of 20–50 nL. At least 5 mN/m surface tension were adjusted by application of a fluorinated surfactant (0.5%-Picosurf1, Sphere Fluidics Limited, Cambridge, UK) [63] in a fluorinated oil (Novec7500, Sphere Fluidics Limited, Cambridge, UK) [54]. Hence, the system operates at low Ca-numbers of about 0.004. On the other hand, the design was adapted to the technical production requirements of the compact disk embossing technology, which was employed to receive appropriate devices. Therefore, two structured masters of mirrored symmetry where prepared by isotropic glass etching of 150 mm glass wafers after microlithographic mask formation. Both glass masters were transferred into nickel embossing tools by electroplating and used in a standard CD (compact disk) production facility. For the channel confined droplet transport of two different droplet sizes (about 35 nL and > 500 nL), two etching depths were required respectively: the main channels were designed with 130 µm depth and the support and dosing channels with 50 µm etching depth. After bonding of both mirrored half-shell discs total channel heights of 260 and 100 µm were received respectively. In contrast to the droplet feeding-channel, the width of the fusion- and outlet-chamber is about 4–9 times larger than their height. Hence, larger droplets which are formed inside the chamber or moved inside the outlet channel have flat “pancake like” geometry (see Figure 2).

The combination of the restricting junctions and the bypass position causes the release of the chamber volume in a self-controlled way. The different cross-sectional areas of the nozzles lead to the self-controlled order of droplet breakthrough. According to the young-la-places law, a droplet will preferably break through a nozzle with the larger cross-sectional area, effected by its characteristic radius. See description in Figure 2. Beside the nozzle geometry the breakthrough of a droplet interface depends on the surface tension. The functionality of this self-controlled volume collection and release unit was combined with electrocoalescence capabilities by the help of two flat metallic electrodes on top of the device (Figure 1h). These electrodes allow the control of coalescence events inside the chamber. 

The developed device offers four different operation modes (see Figure 2):

(1) Self-controlled droplet formation (Figure 2A): The droplet formation is caused by the bypassed chamber layout. In this way, droplets of about 500–2500 nL can be received in a flow rate independent mode.

(2) Single droplet dilution mode (Figure 2B): Single 20–50 nL droplets are merged with a preformed droplet inside the chamber to receive mixed droplets with a volume of approximately 500–2500 nL. In the case of an initial 40 nL droplet a dilution factor of about 1/10 up to 1/70 can be achieved. Proper coalescence is forced by permanent electrocoalescence.

(3) Multiple droplet dilution mode (Figure 2C): Several 20–50 nL droplets were delivered to the chamber and merged with a preformed droplet in the chamber to release a mixed droplet of about 500–2500 nL volume. Electrocoalescence pulses have to be permanently on. 

(4) Multiple droplet coalescence mode (Figure 2D): Here, the chamber release volume was received by collection and subsequent electrocoalescence of multiple 20–50 nL droplets. Droplets of about 500 nL were formed by coalescence of 12 small 40 nL droplets. In this mode no additional dilution solution is required. The electrocoalescence pulses have to be applied at defined time intervals if the chamber is filled completely.

In the case of mode A–C the final droplet volumes and the respective dilution factors can be adjusted. As long as the releasing droplet passes the chamber outlet junction (Figure 1d) the addition of dilution media by the chamber feeding channel (Figure 1h) cause an expansion. In this way, the final droplet volumes can be adjusted within a range of about 500–2500 nL.

The complete microfluidic Disc-device was fixed in a frame (details in Appendix A) on a robotic x–y-table and placed below a microscope camera for optical process monitoring. A homogenous transmissive excitation was realized by the help of a 100 × 100 mm^2^ LED-plate (PHLOX-GC, Phlox-LedRGB-BL 100 × 100) below the device. For fluid actuation, the fluid inlet ports of the device were connected to a precision syringe pump system using a chuck with clamping connectors and 0.5 mm PTFE-tubing [64]. The electrodes were connected to a triggered AC high-voltage power supply system. The overall design and the distance of the electrode were optimized by the help of a simulation (Appendix A). All compounds were connected to a computer systems and program controlled. In this way, camera images, fluidic- and electronic-parameters were recorded and analyzed offline.

## 3. Results and Discussion

The desired functions of the prepared device were investigated and the fluidic operation modes were verified. For the droplet coalescence, the electrical field was applied in form of rectangle pulses of about 450 V with a frequency of about 250 Hz. The voltage could be tuned up to 1250 V. The undefined coalescence of droplets was inhibited by the use of a surfactant (PicoSurfI). The oriented surfactant molecules cause a monomolecular film at the interfaces and stabilize both liquid phases. However, under an electric field the stabilizing effect of the surfactants gets lost and contacting droplets merge.

### 3.1. Self-Controlled Droplet Formation Mode

For the self-controlled droplet formation mode, the separation phase was feed to the chamber consciously through the inlet channel (as defined in Figure 1a), the dispersed-phase was feed continuously from channel (h) (see Figure 3). If the propagating droplet interface reaches the outlet nozzle of the main chamber (d), the droplet propagates into the chamber and the flow of continuous phase is forced through the bypass channel. If the chamber is filled up to the bypass inlet nozzle (e), the bypass flow is blocked by the propagating interface as well. In this state, the pressure-driven flow overcomes the resistance of the main chamber nozzle (d) and the droplet boundary break through this junction. The chamber content is now released (Figure 3c–f). Now the bypass inlet nozzle (e) becomes open again. To bridge this state, a funnel-like channel widening was implemented in the drain channel (c) after the main nozzle. Hence, the releasing droplet expands to a larger radius and seals the bypass outlet nozzle (g) to a certain extend (Figure 3d). The droplet passes through the main nozzle towards the outlet channel (Figure 3e) as long as the chamber is empty and the droplet breaks-off at the nozzle (d) (Figure 3f).

When the continuous phase is supplied at 200 nL/s and the dispersed phase at 50 nL/s, the droplet generator release droplets of about 650 nL with a frequency of about 0.66 Hz. Stable droplet formation was observed up to about 5 Hz at a flow rate ration of about 0.25 with flow rates of 1500 nL/s and 380 nL/s (see Appendix A). However, the volume of the received droplets is defined by the volume of the chamber and the volume flow which is delivered into the droplet by the feeding-channel during the time the droplet needs to pass the chamber outlet nozzle. Consequently, the final volume is defined by the ratio of the continuous phase and the media which is feed through the feeding-channel. In Figure 4 shows the received droplet volume as a function of different flow rate ratio. The volume was determined by video analysis of the droplet formation process and calculated by the 500 nL chamber volume plus the flow rate of the dosing liquid multiplied by the droplet passage time received from the video analysis. Unfortunately, the passage time of the droplet depends very much on the different media parameters such as viscosity and surface tension. Hence, the received droplet volume and formation frequency of has to be determined for all different media and temperature conditions. The present data were received for an aqueous solution of 20 mM Cochenille Red A in deionized water.

### 3.2. Single Droplet Dilution Mode

The single droplet dilution mode was realized by coalescence of a single 40 nL droplet with a pre-formed droplet generated in the chamber by the feeding-channel. The initial (blue) droplets were generated by a T-junction and spaced by a second T-junction feeding the continuous phase (here not shown, Appendix A). For proper one-to-one coalescence, the entering droplet frequency was fine-tuned to about 0.5 Hz by simply adjusting the flow rate of the appropriate media. If the 40 nL droplet enters the chamber, the motion slows down (b) because the driving force of the continuous phase gets lost. The dilution media was delivered through the feeding-channel into the chamber simultaneously. When the dilution media droplet gets in touch with the 40 nL droplet, the applied electrical pulses cause the coalescence, as shown in Figure 5c–e. Hence, the initial droplet (blue) is absorbed in the dilution media if the phase boundaries get in touch. Figure 5 shows the image sequence of the dilution and release process of the droplets. The dilution rate depends on the flow rate ratio of the feeding fluid relative to the carrier fluid, as shown in Figure 4 (see Appendix A).

### 3.3. Multiple Droplet Dilution Mode

However, the dilution is not limited to the coalescence of a single 40 nL droplet. For the multiple droplet coalescence mode, different droplets can be collected inside the chamber and diluted by coalescence with a preformed droplet. By appropriate control of the applied flow rate of continuous-phase at the T-junction in front of the chamber (see Appendix A), the number of droplets entering the chamber can be adjusted. In Figure 6, the coalescence of two initial 25 nL droplets is shown exemplary. The constantly applied electrocoalescence pulses cause the immediate coalescence of approaching interfaces. Depending on the applied flow rate and number of pre-collected droplets in the chamber, dilution processes with a frequency of 0.2–1 Hz were realized. The self-controlled release as well as the constantly applied electrocoalescence pulses enables reproducible and self-controlled dilution processes only at low frequencies. However, the proper control of the delivery frequency of the initial 40 nL droplets into the chamber is a challenging task. Depending on the applied operation mode, dilution rates in a range from about 1:40 (40 nL/1600 nL) to about 1:3 (4 × 40 nL/550 nL) were realized (see Appendix A).

### 3.4. Multiple Droplet Coalescence Mode (D)

An alternative strategy for the volumetric transfer of small droplets is the coalescence with a defined number of droplets of the same size. We made use of this approach to reach a defined volumetric transfer of 40 nL droplets to about 500 nL droplets. For this purpose, a defined number of droplets have to be delivered into the chamber by the fluidic network in front of the coalescence. The dosing channel (Figure 1h) is not required for this operation mode. The main chamber nozzle (d) prevents the outflow of non-merged 40 nL droplets and acts as a barrier in this case. The bypass (e) outbalances the pressure drop between the coalescence chamber and the outlet channel as before. The sequence of filling, collecting and merging of the 40 nL droplets is shown in Figure 7. In any case, the release of the chamber volume (Figure 1a) is initiated when a phase-boundary blocks the bypass nozzle (e) (see Appendix A).

To achieve proper coalescence of all droplets inside the chamber, the interfaces of the individual droplets have to be in contact or close together. The electrocoalescence pulse for droplet merging was controlled by optical inspection and triggered periodically. In contrast to the modes described above, the electrocoalescence pulses have to be applied in a defined time interval. To achieve complete coalescence of all droplets, a series of 10 pulses within 50 ms was applied to induce the coalescence.

The best coalescence results were received if the electrodes were arranged in lateral order on top of the device. Therefore, two flat copper electrodes with a lateral distance of 3 mm were placed above the chamber (Figure 1j). This alignment provides the most efficient coalescence behavior because the droplets are arranged linearly in the electric field. Even in the coalescence mode, an additional diluting fluid may be added through the optional dosing channel (Figure 1f) in order to increase the volume of the unified droplet. However, a stochastically confined mixing of different compartmentalized building blocks can be realized in the same way if droplets with different ingredients are randomly merged.

Almost all biological assays require complex media. To prove the application range for different screening purposes we investigate the influence of salt content, pH-value and viscosity on the electrocoalescence behavior (Figure 8). It was found that neither an enhancement of the salt concentration (NaCl) up to a concentration of 1 M in the aqueous droplets nor the variation of the pH value in the range between 0.2 and 9.5 has any significant effect if a minimal voltage of about 450 V was applied. However, at lower voltages, the coalescence probability decreases at pH 9.5 (adjusted by addition of diluted NaOH) compared to the neutral aqueous solution. The addition of an acid (HCl) had nearly no effect. Significant effects on the coalescence behavior were only found when the applied voltage itself was varied (Figure 8). Whereas a nearly 100% coalescence probability was found at 450 V and above, the coalescence probability decreases to about 90% at 250 V and 50% at 125 V. No coalescence was observed at a voltage of less than 20 V. This characteristic was found for all investigated glycerol concentrations (0 to 20 vol%). At lower voltages, there is a tendency of slightly higher coalescence probability than at higher viscosity. Probably, a higher viscosity reduces the probability of droplet contact during the application of electrical pulses, which leads to the reduced fusion success. However, this effect disappears if a voltage of 450 V or more is applied (more details are given in the Appendix A).

## 4. Conclusions and Outlook

The presented design of a bypassed microfluidic chamber equipped with coalescence electrodes was investigated for four operation modes. The operation of the chamber in the droplet generation mode allows the formation of droplets of about 500–2500 nL volume at about 0.5–5 Hz. The single and multiple-droplet dilution mode dilutes 35–40 nL droplets to a volume of about 500–2500 nL. In this way dilution factors of about 1/104 up to 1/70 were received. The multiple-droplet coalescence mode unifies twelve 40 nL droplets to a mixed droplet. However, the chamber acts as an interface between two different fluidic network dimensions. The first part comprised channel dimensions of about 300 µm for guiding droplets in the range of 30–100 nL, the second 900 × 260 µm channels for guiding 500–2500 nL droplets.

The experimental investigation of the electrocoalescence using different droplet media revealed a broad application range. Typical parameters for biological media such as salt content, pH values and liquid viscosities (glycerol content) were varied. It was found that the coalescence rate is nearly independent on the media composition. Only the applied power has a significant influence. The application of minimum 450 V for the electrocoalescence pulses led to nearly quantitative coalescence rates. The presented chamber layout can be constructively transferred to new fluidic networks which require a volumetric transfer of droplets. We believe even a gentle dimension rescale will not change the fluidic functionality, if the electric field is considered. Especially applications using emulsion-based droplets in the low nL range may need for downstream processing a volumetric expansion. The combination of several drops into one compartment using the multiple-droplet coalescence mode offers new application possibilities by combinatorial mixing strategies. Both examples may give an outlook on future applications of the presented bypassed chamber.

## Figures and Tables

**Figure 1 micromachines-11-00394-f001:**
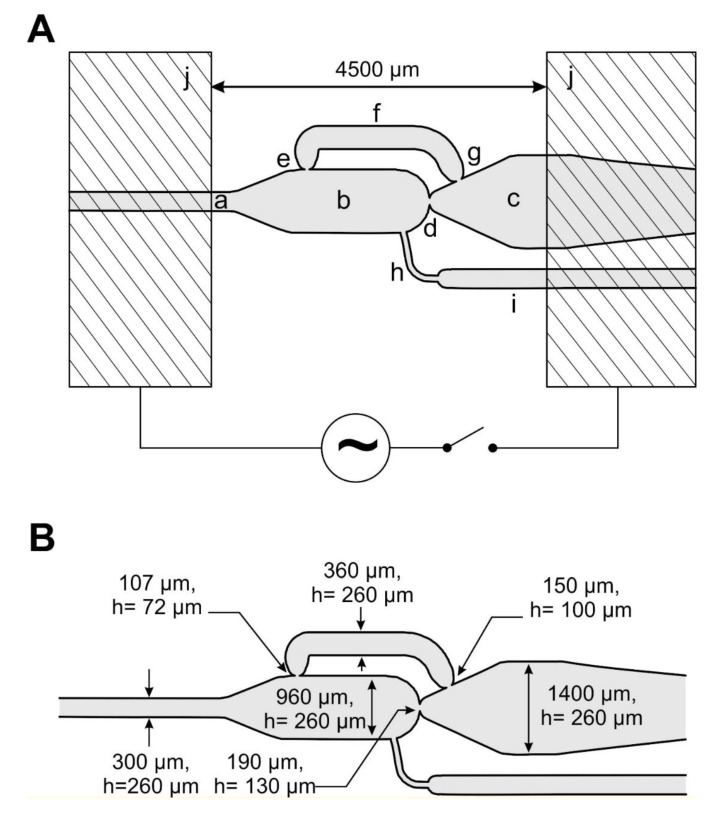
(**A**) Layout of the droplet generation and coalescence device (a) ~40 nL droplet delivery channel, (b) main chamber/coalescence chamber, (c) drain channel for > 500 nL droplets, (d) main channel nozzle, (e) bypass inlet nozzle, (f) bypass channel (g) bypass outlet nozzle, (h) feeding-channel for diluting liquids (dispersed-phase), (i) supply channel, (j) metallic electrode on top of the device. (**B**) Geometric dimensions of the droplet generation and coalescence unit (channel width, h = height).

**Figure 2 micromachines-11-00394-f002:**
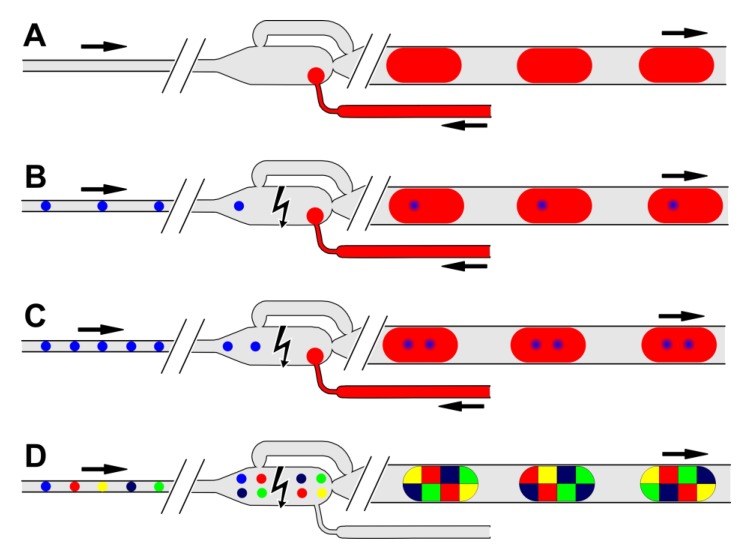
Schematic operation modes of the bypassed chamber: (**A**) Self-controlled droplet formation mode. The dispersed phase (red) is embedded by the carrier phase (gray). Droplet volume is defined by the chamber geometry. (**B**) Single droplet dilution mode. A single droplet (blue) is delivered into the chamber and diluted by coalescence with the diluting media (red). The pulsed electrocoalescence voltage is permanently on. (**C**) Multiple droplets dilution mode. Two—or even more—initial droplets (blue) are delivered into the chamber and diluted by coalescence with the diluting media (red). The pulsed electrocoalescence voltage is permanently on. (**D**) Multiple droplet coalescence mode. The chamber is filled with a defined number of droplets (colored) which are merged by periodically triggered electrocoalescence.

**Figure 3 micromachines-11-00394-f003:**
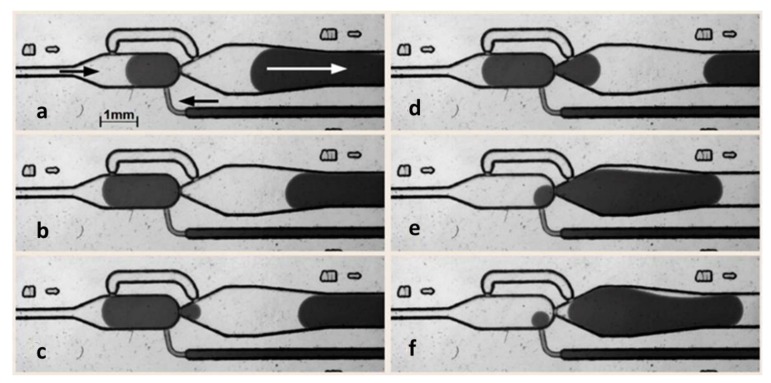
Image series of the self-controlled droplet-formation mode (Figure 2A). The sequence for a repetitive droplet formation goes through the following steps: (**a**) filling of the main chamber, (**b**) sealing of bypass-inlet-junction by the droplets interface, (**c**) breakthrough at the main channel nozzle, (**d**) sealing of bypass-outlet-junction by the droplets interface, (**e**) cast out of droplet with no inflow in the smallest nozzle of bypass-outlet, (**f**) cutting off the droplets at the main chamber nozzle and restarting the sequence. In this operation mode no electrocoalescence is required (details see text).

**Figure 4 micromachines-11-00394-f004:**
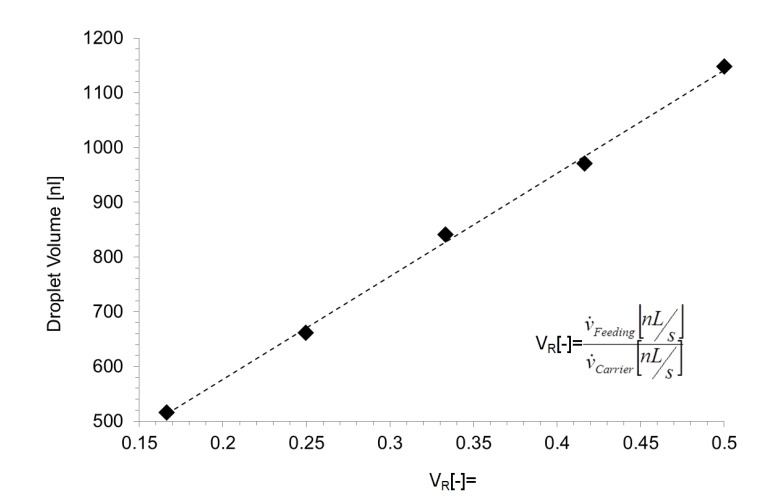
Received droplet volumes for various ratios between feeding- and continuous-phase flowrates V_R_. Feeding-liquid: (dispersed phase), 20 mM aqueous Cochenille Red A and continuous phase: Novec 7500, 0.1 wt% Picosurf. The data received for a total flow rate range of 250–2000 nL/s.

**Figure 5 micromachines-11-00394-f005:**
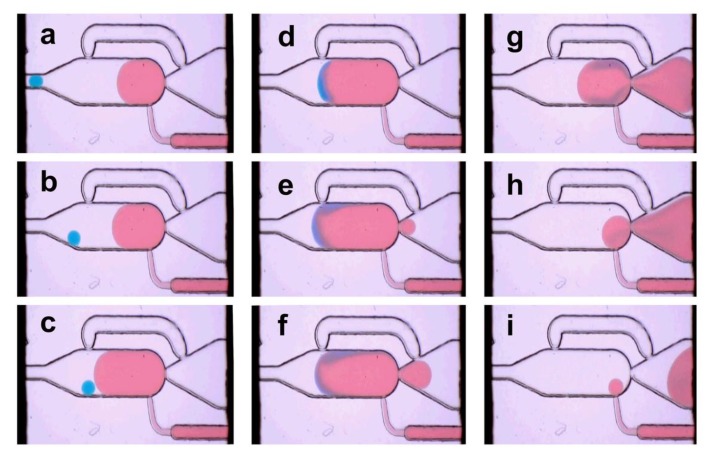
Image series of the single droplet dilution mode (Figure 2B). An initial 40 nL droplet (blue dyed) is delivered into the chamber (**a**,**b**). The dilution media (red dyed) is feed into the chamber in parallel. The electrocoalescence is permanently on and cause the fusion if the boundaries get in touch (**c**,**d**). The expanded droplet of about 550 nL blocks the bypass and initiates the release from the chamber, which is forced by the flow of the carrier medium (**e**–**i**) (frame rate 3 Hz).

**Figure 6 micromachines-11-00394-f006:**
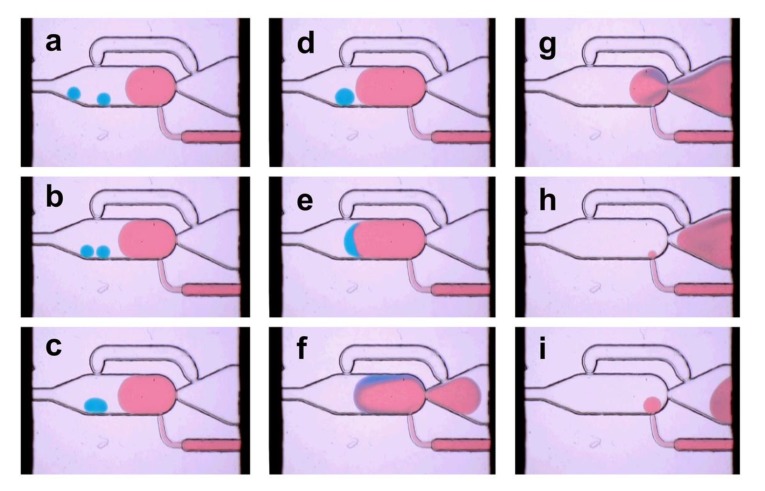
Image series of the multiple-droplet dilution mode (Figure 2C): Two 40 nL droplets (blue dyed) were diluted to about 550 nL (**a**). The dilution media (red dyed) is feed into the chamber in parallel. The electrocoalescence is permanently on and cause the fusion if the boundaries get in touch (**b**,**c**) and (**d**,**e**). The expanded droplet blocks the bypass and initiates the release from the chamber, which is forced by the flow of the carrier medium (**e**–**i**) (frame rate 3 Hz).

**Figure 7 micromachines-11-00394-f007:**
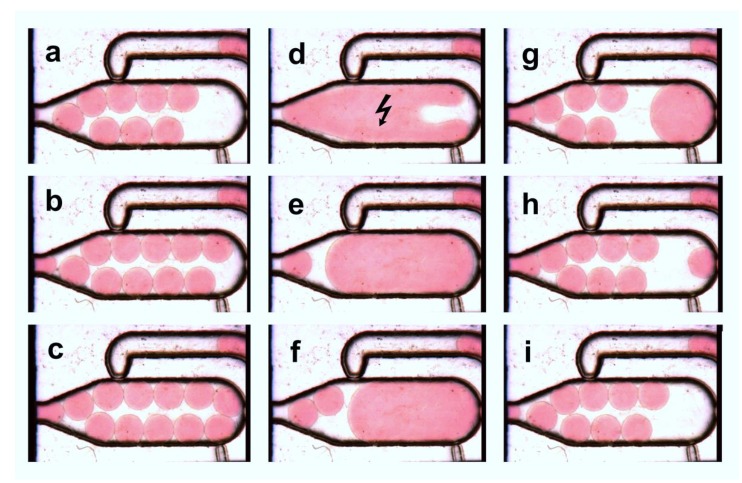
Image series of multiple-droplet coalescence mode (Figure 2D): Always 12 droplets (red dyed) of about 40 nL are delivered into the chamber (**a**–**c**). The main chamber nozzle acts as a barrier for the droplets. When the chamber is filled a defined electrocoalescence pulse merge all droplets (**d**). The blocked bypass initiates the droplet release from the chamber, which is forced by the feeding carrier media (**e**). As the coalesced droplet passes through the main chamber nozzle, new small droplets refill the chamber (**e**–**i**).

**Figure 8 micromachines-11-00394-f008:**
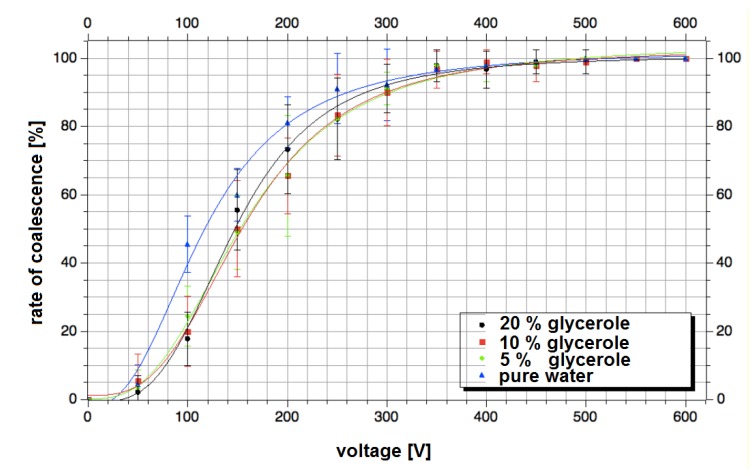
Coalescence rate in dependence of applied electrocoalescence voltage for different media. Water droplets with different glycerol content were delivered into the chamber. The electrocoalescence pulse interval was adjusted to the time need to fill the chamber with droplets. The number of coalesced droplets was determined by video analysis. The average coalescence rate was calculated for 100 repetitive coalescence events per spot.

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
