# Peer review of "Microfluidic Chamber Design for Controlled Droplet Expansion and Coalescence"

_micromachines, 2020, doi:10.3390/mi11040394_

Round 1
Reviewer 1 Report
Authors have presented a droplet processing network in the form of microfluidic channels that are interfaced with the voltage inputs.
The modularity of the microfluidic design is interesting and can be interesting to generate various combinations of droplets.
However, the figures and figure captions need significant revisions. It needs more clarity and editing. Figures need labels and better organization. Figure captions need to be longer, describing what are shown on the each part of the Figure. Four unique functions in this microfluidic interface should be carefully presented in the Figures.
Author Response
Dear Reviewer,
thank you for reviewing our manuscript and helpful suggestions. We revised the manuscript following your suggestions. All revisions in the manuscript were highlighted. Please consider our response on the other reviewer suggestions. Thank you for helping to improve our manuscript. We hope that the revised version can be published. If you have any other concerns please do not hesitate to contact us again.
Comments and Suggestions for Authors
However, the figures and figure captions need significant revisions. It needs more clarity and editing. Figures need labels and better organization. Figure captions need to be longer, describing what are shown on the each part of the Figure.
The figure captions were revised. The details of the depicted processes are described in the main text. Hence, we did not want to give the description redundantly. Some figures were moved to the SI.
Four unique functions in this microfluidic interface should be carefully presented in the Figures.
We add a new scheme (figure 2) to the manuscript showing the descried four operation modes.

Reviewer 2 Report
Major comments:
1) The title is hard to understand. I suggest simplifying a lot: e.g. Microfluidic chamber for controlled mixing and coalescence, or something similar
2) In the abstract and the main text, authors should decide if their work is about "the device" or about "the chamber" and then pick one. Currently it is very hard to follow the text and the point of the paper (which I presume is the design of the chamber) goes missing
3) Language: I suggest having native speaker scientist to look at the text as it is quite difficult to follow sometimes. Sentences are too long and complicated. I suggest strongly to simplify and shorten the sentences.
4) It is currently very hard to understand The Need for such device. Authors should add such strong sentence to the abstract and maybe a short paragraph to intro, that describes strongly why such dilution unit is needed. Plus add some examples where it could be useful
5) I also miss a separate intro section about microfluidic dilutors, how different groups have handled this issue before
6) Fig 1 should add a general schematic of the example for the whole fluidic system (or droplet processing network) that includes all components (big droplet formation, small droplet formation etc). Then it is better to understand the whole idea. For example, simplified version of SI, Fig1. Then explain 4 operation modes on the schematic
7) The main text, especially results and discussion feels too long. Suggest shortening it and moving some parts to SI. For example shorten the 3.4 and include 1-2 images max, rest can go to SI
8) Conclusion should be simplified and shortened, such long, repetitive and detailed description feels like deja vu of the results. Concentrate of the key findings only. But still add 1-2 sentences about the potential applications of the chamber.
Minor comments:
1) typo in the abstract, line 6. "cannel"
2) How transferable is their chamber design for other segmented flow based designs?
3) Fig 6 and 7 are too long. 8-10 pics per operation should be enough. Merge the images afterwards
4) Fig 9 and corresponding text looks like material for SI. One sentence in the main text is enough
5) Authors should describe briefly how their invention works in the context of their application. I presume it's bacterial screening based on their intro. How does the dilutor-coalescence chip help their grand plan?
Author Response
Dear Reviewer,
thank you for reviewing our manuscript and detailed suggestions. We revised the manuscript following your suggestions. All revisions in the manuscript were highlighted. Please consider our response on the other reviewer suggestions. Thank you for helping to improve our manuscript. We hope that the revised version can be published. If you have any other concerns please do not hesitate to contact us again.
Major comments:
1) The title is hard to understand. I suggest simplifying a lot: e.g. Microfluidic chamber for controlled mixing and coalescence, or something similar
You are right. We changed the title to “Microfluidic Chamber Design for Controlled Droplet Expansion and Coalescence”
2) In the abstract and the main text, authors should decide if their work is about "the device" or about "the chamber" and then pick one. Currently it is very hard to follow the text and the point of the paper (which I presume is the design of the chamber) goes missing
The abstract was revised following your suggestions. We omit the word device.
3) Language: I suggest having native speaker scientist to look at the text as it is quite difficult to follo w sometimes. Sentences are too long and complicated. I suggest strongly to simplify and shorten the sentences.
Proofreading was done by skilled scientist again. We tried our best to improve the comprehensibility of the overall text.
4) It is currently very hard to understand The Need for such device. Authors should add such strong sentence to the abstract and maybe a short paragraph to intro, that describes strongly why such dilution unit is needed. Plus add some examples where it could be useful
We add in the abstract as well as the introduction appropriate explanations focusing on the “concentration gradient” aspect. However, intention is the description of the chamber design. Readers which are skilled in the design of microfluidic devices can hopefully benefit and integrate the design in their own applications.
5) I also miss a separate intro section about microfluidic dilutors, how different groups have handled this issue before
Beside droplet generation the volumetric expansion is one of the basic operations in droplet based microfluidics. It is described in many of the papers we cite. However, we add in the introduction a section and some more specific publications [11,16, 35-37].
6) Fig 1 should add a general schematic of the example for the whole fluidic system (or droplet processing network) that includes all components (big droplet formation, small droplet formation etc). Then it is better to understand the whole idea. For example, simplified version of SI, Fig1. Then explain 4 operation modes on the schematic
We add a Figure 2 explaining the four operation modes schematically. But, we did not add an overall network scheme. In our modest opinion it would requires more detailed and additional explanations and would go beyond the scope of this paper.
7) The main text, especially results and discussion feels too long. Suggest shortening it and moving some parts to SI. For example shorten the 3.4 and include 1-2 images max, rest can go to SI
You are right. We moved two more figures and the belonging text to the SI.
8) Conclusion should be simplified and shortened, such long, repetitive and detailed description feels like deja vu of the results. Concentrate of the key findings only. But still add 1-2 sentences about the potential applications of the chamber.
We changed the headline to “conclusions and outlook”. The overall paragraph was revised completely. Two examples for future applications were given as outlook.
Minor comments:
1) typo in the abstract, line 6. "cannel"
Typo was checked again.
2) How transferable is their chamber design for other segmented flow based designs?
We believe an experienced fluidic designer can extract all required layout information from Figure 1. We have no doubt that the design can be transferred in other material systems or scaled for about +- 20% size.
3) Fig 6 and 7 are too long. 8-10 pics per operation should be enough. Merge the images afterwards
You are right the information is “dense”. We hope the image series will help even non-expert readers to understand the depicted processes clearly.
4) Fig 9 and corresponding text looks like material for SI. One sentence in the main text is enough
Figure 9 and 10 were moved to SI
5) Authors should describe briefly how their invention works in the context of their application. I presume it's bacterial screening based on their intro. How does the dilutor-coalescence chip help their grand plan?
We add some more explanations to the introduction. Unfortunately, more detail of the overall screening approach will go beyond the scope of this paper.

Round 2
Reviewer 2 Report
Dear authors,
Thank you for responding quickly to suggestions.
Major comment:
My overall suggestion still remains that the paper would win if the results section is shortened, especially the figures. For me it makes more sense if i) Figures 5, 6 and 7 are merged together into single image, ii) each having just 6-8 images describing the process, iii) fig 6-8 captions should explain the processes a bit more in details
Minor comment:
Fig2 is missing the caption
Author Response
Dear Reviewer,
Thank you for helping us improving our manuscript with your helpful suggestions. We followed your suggestion and reduced the number of pictures in figures 5, 6 and 7. Even the captions were supplemented to explain the content in greater detail. If you have any further concerns, please do not hesitate to contact us again.
Sincerely yours,
A. Groß